# Anterior Segment Parameter Changes after Cataract Surgery in Open-Angle and Angle-Closure Eyes: A Prospective Study

**DOI:** 10.3390/jcm12010327

**Published:** 2022-12-31

**Authors:** Kangyi Yang, Zhiqiao Liang, Kun Lv, Yao Ma, Xianru Hou, Huijuan Wu

**Affiliations:** 1Department of Ophthalmology, Peking University People’s Hospital, Beijing 100044, China; 2Eye Diseases and Optometry Institute, Beijing 100044, China; 3Beijing Key Laboratory of Diagnosis and Therapy of Retinal and Choroid Diseases, Beijing 100044, China; 4College of Optometry, Peking University Health Science Center, Beijing 100044, China

**Keywords:** primary angle-closure glaucoma, anterior rotation of ciliary processes, mechanisms, cataract surgery, ultrasound biomicroscopy

## Abstract

Background: To investigate the anterior segment parameters before and after cataract surgery in open-angle eyes and different subtypes of primary angle-closure glaucoma (PACG) eyes and to further explore the potential relationship between the anterior rotation of the ciliary process and crystalline lens. Methods: An observational, prospective study was performed on 66 patients who had cataract surgery including 22 chronic PACG patients, 22 acute PACG patients, and 22 open-angle cataract patients. Anterior segment parameters including the trabecular-ciliary process distance, ciliary process area, trabecular-ciliary angle (TCA), maximum ciliary body thickness (CBTmax), and so on, were measured using ultrasound biomicroscopy preoperatively and 3 months postoperatively. Results: After the surgery, there were significant increases in TCA (*p* < 0.001) and CBTmax (*p* < 0.05) in all three groups, while there was no significant change in the trabecular-ciliary process distance (*p* > 0.05) in all three groups. No significant difference in the changes of ciliary process area, TCA, and CBTmax (*p* > 0.05) pre- and postoperatively among the three groups were identified. Conclusions: Extractions of crystalline lenses played similar roles in terms of decreasing the anterior rotation of ciliary processes in open-angle eyes and angle-closure eyes. A natural anatomical abnormality may be a more important factor in the anterior rotation of ciliary processes in PACG patients.

## 1. Introduction

Primary angle-closure glaucoma (PACG) is characterized by elevated intraocular pressure (IOP) and obstruction of aqueous outflow as a result of mechanical obstruction of the trabecular meshwork, which is one of the leading causes of global irreversible blindness [1,2]. PACG can be classified into acute PACG and chronic PACG according to the characteristics of clinical presentation [3]. During the acute attack stage of acute PACG, the iris quickly and completely occludes the entire trabecular meshwork, and by contrast, in chronic PACG, the iris slowly creeps over the trabecular meshwork [3].

The majority of PACG cases in China result from a combination of multiple mechanisms, including pupillary block factors and non-pupillary block factors, such as a thick peripheral iris, a plateau iris, anterior attachment and insertion of the iris root, a forward shift of the lens, and anterior rotation of the ciliary processes exist [4]. Quantitative measurements revealed that the characteristics of the ciliary processes are one of the most important differences between eyes with angle-closure and normal eyes [3]. The ciliary processes of angle-closure eyes rotated more anteriorly than that of open-angle eyes. Additionally, the crystalline lens was considered to be involved in the pathogenesis of PACG because of the continuous growth in lens volume and lens vault (LV) [5,6]. The relationship between the anterior rotation of ciliary processes and the crystalline lens was also discussed in previous studies. In some studies, it had been reported that cataract surgery can widen the anterior chamber angle not only by decreasing lens volume and relieving pupillary block, but also by attenuating anterior positioning of the ciliary processes in the eyes with PACG [5,7]. Man et al. [8] even demonstrated that after lens extraction and intraocular lens (IOL) implantation, all PACG eyes had significantly longer trabecular-ciliary process distance (TCPD) and deeper aqueous depth (AQD). On the other hand, Li et al. [9] found that although there was a significant increase in AQD in acute primary angle-closure patients after phacoemulsification, the difference in TCPD was not significant. Therefore, whether the anterior traction of the lens to ciliary processes plays a role in the anterior rotation of ciliary processes or not remains ambiguous.

In this study, ultrasound biomicroscopy (UBM) had been used to evaluate anterior segment parameters before and after phacoemulsification and IOL implantation in acute PACG, chronic PACG, and open-angle eyes and to further explore whether the anterior rotation of ciliary processes has a potential relationship with the crystalline lens or not, which may provide more detailed information related to the pathogenesis of PACG.

## 2. Materials and Methods

### 2.1. Study Design

This was an observational, prospective study of Chinese subjects approved by the Ethics Committee of Peking University People’s Hospital (approval number 2022PHB256-001) and it followed the tenets of the Declaration of Helsinki. Written informed consent was obtained from all subjects prior to recruitment.

### 2.2. Patients

Subjects diagnosed with PACG were recruited from the glaucoma clinics of Peking University People’s Hospital from June 2016 to August 2022 and were classified as acute PACG and chronic PACG. Patients who visited clinics due to age-related cataracts were enrolled as the control group and were characterized by an open anterior chamber angle. Accordingly, 68 eyes of 68 subjects were included, of which 2 eyes were excluded due to poor UBM image quality.

Eyes with acute PACG were defined as eyes with two of the following symptoms: ocular or periocular pain, headache, nausea and/or vomiting, blurred vision, and halos around lights; and the following ophthalmologic findings: an IOP of more than 30 mm Hg, conjunctival hyperemia, corneal epithelial edema, a shallow anterior chamber with angle-closure, iris bombe and mid-dilated pupil, along with glaucomatous optic neuropathy and visual field defect.

Eyes with chronic PACG were defined as eyes without symptoms or signs of a prior acute attack, including glaucomatous fleck, keratic precipitates, or iris atrophy. Patients had more than three cumulative hours of peripheral anterior synechia and a chronically elevated IOP (>21 mm Hg), along with glaucomatous optic neuropathy or visual field defect.

Exclusion criteria included (1) secondary angle closure such as iris neovascularization, trauma, tumor, uveitis, and lens subluxation, (2) prior intraocular surgery, such as cataract surgery, trabeculoplasty, and trabeculotomy, (3) inability to tolerate gonioscopy or UBM examinations, and (4) cataract patients with nuclear opalescence, nuclear color or cortical cataract of more than grade 3 (LOCS III classification).

### 2.3. Ophthalmologic Examinations

All patients underwent ophthalmologic examinations, including best corrected visual acuity (BCVA) measured by the Snellen visual chart, IOP measurement by Goldmann applanation tonometry (Haag-Streit, Koniz, Switzerland), and axial length measured by IOLMaster biometry (IOLMaster 700, Carl Zeiss Meditec, 137 Inc., Dublin, CA, USA). Gonioscopy was performed in a dimly lit room by a glaucoma specialist using a Zeiss-style four-mirror gonioscopy lens (Model G-4, Volk Optical, 154 Inc., Mentor, OH, USA) at 16× magnification with and without indentation. Optical coherence tomography (Spectralis HRA + OCT, Heidelberg Engineering, Heidelberg, Germany) was used to test retinal nerve fiber layer defect and a visual field test (Humphrey Field Analyzer, Carl Zeiss Meditec, Inc., Dublin, CA, USA) was performed to show characteristic glaucomatous visual field defects.

### 2.4. Ultrasound Biomicroscopy

The anterior chamber configuration was determined before and at 3 months after cataract surgery using UBM (Aviso, Quantel Medical, Inc., Bozeman, MT, USA) and a 50-MHz transducer probe by an experienced operator (Y.W.) who was masked to the clinical data. Examinations were performed under the same room illumination, with the patient in the supine position. After topical anesthesia by using 0.4% oxybuprocaine hydrochloride (Benoxil^®^; Santen, Tokyo, Japan) eye drops, a plastic eye cup containing physiologic saline was mounted on the globe. To minimize accommodation, patients were instructed to fixate with the contralateral eye on a distant ceiling target. Each subject’s eyes were measured in the superior, inferior, temporal, and nasal quadrants, as well as nasal and temporal scans centered on the pupil to obtain complete images of the anterior segment. Analyses were limited to images that clearly showed the scleral spur (SS), iris, ciliary body, and anterior surface of the lens.

### 2.5. Image Analysis

The UBM Images of chronic PACG, acute PACG, and open-angle cataracts were measured quantitatively in all four quadrants using an in-built caliper in the UBM software by a single examiner masked to clinical data. The SS was identified based on the differential tissue density between the collagen fibers of the SS and the longitudinal muscle of the ciliary body as a crucial anatomical mark in the curvature of the inner surface of the angled wall. The anterior segment parameters [10,11,12,13,14] were measured according to the methods of our previous publication and other research. The parameters on full-view scans at the nasal-temporal position were as follows. (1) AQD: the distance between the posterior surface of the central cornea and the anterior surface of the crystalline lens or IOL in the midline of the pupil. (2) Pupil diameter (PD): the shortest distance between the pupil edges of the iris cross-sections. (3) LV: the perpendicular distance from the anterior pole of the lens to the horizontal line between the SS. (4) Anterior chamber width (ACW): the distance between the two SS’s. (5) Iris area: the cumulative cross-sectional area of the full length (from spur to pupil) of the iris. (6) Ciliary sulcus diameter (CSD): the distance between the two horizontal diameters for the ciliary sulcus (Figure 1). Parameters [10,11,12,14] measured on the radial scans at the superior, nasal, inferior, and temporal positions were as follows. (1) An iris thickness of 750 µm (IT 750) and 2000 µm (IT 2000): an iris thickness of 750 µm and 2000 µm from the SS. (2) An angle-opening distance of 500 µm (AOD 500) and 750 µm (AOD 750): the distance between the posterior corneal surface and the anterior iris surface on a line perpendicular to the trabecular meshwork at 500 µm and 750 µm from the SS. (3) A trabecular iris space area of 500 µm (TISA 500) and 750 µm (TISA 750): the area bounded anteriorly by AOD 500 and AOD 750, as determined posteriorly by a line drawn from the SS perpendicular to the plane of the inner scleral wall to the iris, superiorly by the inner corneoscleral wall, and inferiorly by the iris surface. (4) A trabecular iris angle of 500 µm (TIA 500) and 750 µm (TIA 750): the apex of the angle at the iris recess and the arms of the angle passing through a point on the trabecular meshwork at 500 µm and 750 µm from the SS and the point on the iris perpendicularly opposite. (5) A ciliary process area (CPA): the cross-sectional area of the ciliary process bounded laterally by a line connecting the insertion location of the iris into the ciliary body and the crosspoint of a line of 500 µm from the SS perpendicular to the plane of the inner scleral wall to the ciliary process, and internally by the ciliary process surface. (6) A trabecular-ciliary angle (TCA): the angle between the posterior corneal surface and the anterior surface of the ciliary body. (7) TCPD: the length of the line extending from the corneal endothelium 500 mm from the SS perpendicularly through the posterior surface of the iris to the ciliary process. (8) The maximum ciliary body thickness (CBTmax): the thickest location of the ciliary body (Figure 2). The vitreous zonule (VZ) [13]: the bridging bundles of zonular fibers running from the region of the zonular plexus in the valleys of the posterior pars plicata toward the vitreous membrane in the region of the ora serrata (Figure 3).

### 2.6. Surgical Technique

All phacoemulsification and hydrophobic acrylic TECNIS ZA9003 IOL (Abbott Medical Optics Inc., Santa Ana, CA, USA.) implantation surgeries were performed by the same experienced surgeon (H.W.) and were then followed up by two ophthalmologists (Z.L. and H.W.). The pupil was dilated six times using 0.5% tropicamide phenylephrine (Santen, Osaka, Japan) 30 min prior to surgery. Phacoemulsification was performed under topical anesthesia using 0.4% oxybuprocaine hydrochloride. The corneal incision was performed at 11 o’clock. The chamber was immediately deepened using 1.7% sodium hyaluronate (Bausch & Lomb, Jinan, China). After continuous curvilinear capsulorhexis, the nucleus was removed with no intraoperative complications. An automated irrigation/aspiration apparatus was introduced into the anterior chamber to remove the cortical remnants and polish the posterior lens capsule. The intra-IOL was placed in the capsular bag. Postoperative treatment consisted of 0.5% tropicamide phenylephrine (Mydrin^®^-P; Santen, Osaka, Japan) administered once per night for 1 month, tobramycin and dexamethasone eye drops (Tobradex^®^; Alcon, Belgium) four times a day for 1 month, and 0.5% levofloxacin (Cravit^®^; Santen, Osaka, Japan) eye drops four times a day for 1 week after surgery.

### 2.7. Repeatability and Reproducibility

We performed repeatability and reproducibility analysis of UBM parameters. Five patients in each group were randomly selected for analysis. The first observer (K.Y.) measured parameters twice within two weeks to test intra-observer variability. A second observer (L.K.) measured the same images independently on a different day to decide inter-observer variability. The intra-observer and inter-observer variability were calculated using the coefficient of the intra-class coefficients.

### 2.8. Statistical Analysis

The results were analyzed using SPSS 22.0 (SPSS, Inc., Chicago, IL, USA). All data were calculated as a mean ± standard deviation. ANOVA tests or Kruskal–Wallis tests were used to compare the continuous variables (age, BCVA, IOP, axial length, AQD, PD, LV, ACW, CSD, iris area, IT 750/2000, AOD 500/750, TISA 500/750, TIA 500/750, CPA, TCA, TCPD, CBTmax, and the number of quadrants of VZ). The Shapiro–Wilk test was used to test whether the variables conform to normal distribution. The test showed that the data (age, CPA, TCPD, AQD, CSD, ACW, LV, IT 750, and iris area) conformed to normal distribution; therefore, analysis of Variance (ANOVA) tests were used to compare parametric variables. Alternatively, Kruskal–Wallis test, in the case of non-normally distributed variables, was used to compare non-parametric variables (BCVA, IOP, axial length, TCA, CBTmax, PD, IT 2000, AOD 500/750, TIA 500/750, TISA 500/750, and the number of quadrants of VZ). As for variance comparison among groups, the least significant difference (LSD) test was used for homogeneity, while the Tamhane test was used for heterogeneity. χ2 tests were used to compare the categorical variables (gender) among the three groups. Paired contrast *t*-tests were used for the preoperative and postoperative comparison of AQD, PD, LV, ACW, CSD, iris area, IT 750/2000, AOD 500/750, TISA 500/750, TIA 500/750, CPA, TCA, TCPD, CBTmax, and the number of quadrants of VZ. A *p*-value of less than 0.05 was considered statistically significant.

## 3. Results

### 3.1. Demographics, Clinical Characteristics, and Preoperative Parameter Data

Sixty-six eyes of 66 patients including 22 chronic PACG eyes, 22 acute PACG eyes, and 22 open-angle cataract eyes were enrolled in this study. The demographic, clinical characteristics, and preoperative anterior segment UBM parameters are shown in Table 1. Among all participants, the mean (± SD) age was 69.42 ± 7.82 years. Chronic PACG and acute PACG eyes had higher IOP and smaller axial length than open-angle eyes (*p* < 0.05). There was no significant difference in age, BCVA, and gender among the three groups (*p* > 0.05) (Table 1). Compared to open-angle eyes, chronic PACG and acute PACG eyes had smaller TCA, TCPD, CBTmax, AQD, ACW, TIA 500, TIA 750, AOD 500, AOD 750, TISA500, TISA 750, IT 750, a fewer number of quadrants of VZ, and larger CPA and LV preoperatively (*p* < 0.05). Chronic PACG eyes had larger AQD, and smaller LV and PD than acute PACG patients (*p* < 0.05). Acute PACG eyes had larger PD and a smaller iris area than open-angle cataract eyes (*p* < 0.05). No significant differences were found regarding CSD and IT 2000 preoperatively (*p* > 0.05).

### 3.2. Postoperative Parameter Data

There was a significant increase in TCA, CBTmax, AQD, LV, TIA 500, TIA 750, AOD 500, AOD 750, TISA 500, and TISA 750 (*p* < 0.05), while the number of quadrants of VZ was significantly decreased (*p* < 0.05) in all three groups 3 months after surgery. ACW, IT 750, and IT 2000 were significantly increased in chronic PACG eyes and acute PACG eyes postoperatively (*p* < 0.05). Compared to preoperative parameters, acute PACG eyes had smaller CPA, CSD, and a larger iris area (*p* < 0.05) postoperatively. Preoperative open-angle cataract eyes had smaller PD and a larger iris area 3 months after surgery (*p* < 0.05). No significant differences were found regarding TCPD in all three groups, respectively, 3 months after surgery (*p* > 0.05).

The comparisons of the changes in anterior segment parameters after cataract surgery among chronic PACG, acute PACG, and open-angle cataract groups are shown in Table 2. The extent of increase in AQD, LV, and IT 2000 was significantly larger in acute PACG eyes than in chronic PACG and open-angle cataract eyes after surgery (*p* < 0.05). Compared to open-angle cataract eyes, chronic PACG and acute PACG eyes had a larger increase in IT 750 postoperatively (*p* < 0.05). There was no significant difference in change in the CPA, TCA, TCPD, CBTmax, CSD, ACW, PD, TIA 500, TIA 750, AOD 500, AOD 750, TISA 500, TISA 750, iris area, and the number of quadrants of VZ among the three groups before and after surgery (*p* > 0.05).

### 3.3. Intra-Class Coefficient Data

The intra-observer and inter-observer intra-class coefficients were 0.869–0.999 and 0.852–0.997, respectively (Table 3), which showed good repeatability and reproducibility of UBM parameters measured in this study.

## 4. Discussion

In this study, the anterior segment parameters before and after phacoemulsification and IOL implantation were investigated in open-angle eyes and different subtypes of PACG eyes. There was no significant difference identified between the three groups in the change of degree of anterior rotation of the ciliary body before and after cataract surgery. The effect of anterior traction by the crystalline lens on the anterior rotation of ciliary processes might be equal in open-angle eyes and angle-closure eyes. Hence, a natural anatomical structure abnormality might be another pathogenesis for anterior rotation of ciliary processes in PACG patients. To our knowledge, this is the first study to explore whether the anterior rotation of ciliary processes has a potential relationship with the crystalline lens or not in the pathogenesis of PACG by comparing the parameters in open-angle and angle-closure patients before and after phacoemulsification and IOL implantation.

Despite the increasing number of studies focusing on ciliary processes and crystalline lenses about their role in the pathogenesis of PACG recently [7,8,9,12,15,16], the relationship between them remains ambiguous. He et al. [14] proposed that healthy Chinese had smaller TCPD, TCA, and a thinner ciliary body thickness than healthy Caucasians. They suggested that the Chinese have a more anteriorly positioned lens, which may contribute to more anteriorly positioned ciliary processes and the thinner ciliary body might cause anterior positioning and thickening of the lens through the greater anterior positioning of the ciliary processes and the loosening of the zonules [14]. In the current study, the CBTmax of PACG groups was significantly thinner than the open-angle group, which is consistent with a previous study [14]. However, the results from our study demonstrated the extractions of crystalline lenses played similar roles in terms of decreasing the anterior rotation of ciliary processes in open-angle eyes and angle-closure eyes, which suggested that the anterior rotation of ciliary processes might be a natural anatomical structure in PACG patients rather than due to a lens-related reason. Ünsal et al. [12] found TCPD and CBTmax increased at 2 months after surgery in 36 cataract eyes. However, the difference did not reach significance. Pereira et al. [17] suggested that TCPD did not change after cataract surgery in eyes with an open-angle. Li et al. [9] also found TCPD increased with no significant difference in acute primary angle-closure eyes after phacoemulsification, and suggested that the lack of significant change in TCPD after surgery may be due to a small change in the position of the ciliary process as a result of cataract extraction. However, Nonaka et al. [7] described TCPD increased significantly after cataract surgery in eyes with primary angle-closure or PACG and proposed that lens extraction can reposition the ciliary processes in a more posterior location. Unfortunately, the above studies did not compare patients with open-angle and closed-angle in the same study. Moreover, in contrast to previous studies demonstrating that TCPD increased dramatically in PACG eyes that received lens extraction [8,18], the result of the current study was that there was no significant TCPD increase in all groups postoperatively, which did not support the hypothesis of anterior rotation of ciliary processes resulting from anterior traction of the crystalline lens. In addition to TCPD, TCA could also reflect the anterior positioning of the ciliary processes. In the current study, the changes in TCA and TCPD among open-angle eyes and different subtypes of PACG eyes were not statistically significant postoperatively, which indicated that the attenuate effects of lens extraction were consistent in all three groups. Compared to open-angle eyes, PACG eyes maintained a narrower TCA and smaller TCPD after cataract surgery, indicating that anteriorly rotated ciliary processes was not completely eliminated by lens extraction in the PACG eyes. Therefore, the forward traction of crystalline lens can lead to the anterior rotation of the ciliary processes, but is not the only factor.

Previous studies demonstrated that the size of the anterior ciliary body might be related to the crystalline lens [16]. The ciliary muscle contraction caused forward and inward rotation of the ciliary body and increased the thickness of the anterior ciliary body [19]. Theoretically, if the anterior rotation of ciliary processes in patients with PACG was due to the crystalline lens anterior pulling of ciliary processes, CPA would decrease after cataract extraction. However, in the current study, there were no significant differences in the changes of CPA after cataract surgery among the three groups, which was the evidence to show that lens removal did not narrow the gap in the size of the anterior ciliary body between closed- and open-angle eyes in terms of CPA. Therefore, the forward traction of the crystalline lens has little effect on the shape of the ciliary processes in both PACG eyes and open-angle eyes.

An experimental study found after VZ lysis, the ciliary body moved forward which narrowed the anterior chamber angle [13]. Shon et al. [20] described the relationship between VZ and anterior chamber angle characteristics in two clusters of PAC and PACG eyes and demonstrated that eyes with no VZ appeared to have smaller TCA, TCPD, AOD 500, AOD 750, TISA 500, and TISA 750 than eyes with visible VZ. In the current study, chronic PACG eyes and acute PACG eyes had a smaller number of quadrants of VZ, narrower angle, and more anterior rotation of ciliary processes than open-angle eyes before surgery. Anatomically, PACG eyes lacking VZ may play a potential role in angle closure. Weakened zonular fibers may rupture during surgery [21], which could explain the significant decrease in VZ in all three groups after surgery. Thus, the loss of VZ may have counterbalanced the posterior movement of the ciliary process caused by cataract extraction. Hence, further explorations about VZ and cataract extraction with larger sample sizes are warranted.

Modesti et al. [22] proposed that there was a decrease in the ciliary ring diameter and CSD, as well as changes in the shape of ciliary processes, resulting from the absence of zonular fiber tension after lens extraction. In the current study, acute PACG eyes had smaller CSD postoperatively. Though the changes in CSD among open-angle eyes and different subtypes of PACG eyes were not statistically significant postoperatively, chronic PACG and acute PACG eyes had smaller postoperative CSD than open-angle eyes exposed persistent anterior rotation of ciliary processes in PACG eyes. Further research is needed to investigate the correlation between CSD and ciliary processes in PACG eyes.

In the current study, we showed that as a natural anatomical abnormality, the anterior rotation of ciliary processes in PACG patients may not be associated with the lens factor. It is important to increase our understanding of the pathogenesis of PACG, to urge us to further explore the cause of anterior rotation of ciliary processes, such as the association with the presence of vitreous zonule. Reducing the volume of the ciliary body or relieving the anterior rotation of ciliary processes may further help us to carry out new treatments for PACG patients, such as cyclophotocoagulation.

Several limitations need to be noted in the present study. Firstly, the sample size was relatively small and there was no comparison with another imaging modality. Secondly, although miotics and mydriatics were not used before UBM examination, some angle-closure eyes used other anti-glaucoma drugs before cataract surgery. The potential effect of these drugs on anterior segment structure is not clear. Finally, since all patients had some degree of cataracts, we are unable to explore the potential relationship between the ciliary body and transparent lens at the moment.

## 5. Conclusions

This study demonstrates that extractions of crystalline lenses played similar roles in terms of decreasing the anterior rotation of ciliary processes in open-angle eyes and angle-closure eyes, and the anterior rotation of ciliary processes could not be eradicated completely postoperatively in angle-closure eyes. More attention should be paid to the natural anatomical abnormality of anterior rotation of ciliary processes, as it may provide more detailed information related to the pathogenesis of PACG.

## Figures and Tables

**Figure 1 jcm-12-00327-f001:**
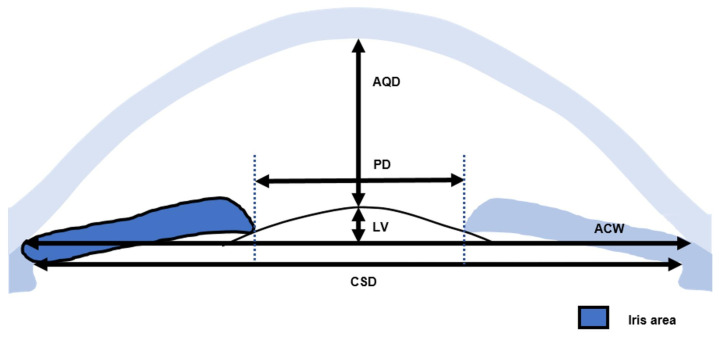
Anterior segment parameters on full-view scans at the nasal-temporal position measured by ultrasound biomicroscopy (UBM). (Illustrator: K.Y.). AQD: the distance between the posterior surface of the central cornea and the anterior surface of the crystalline lens or IOL in the midline of the pupil; PD: the shortest distance between the pupil edges of the iris cross-sections; LV: the perpendicular distance from the anterior pole of the lens to the horizontal line between the SS; ACW: the distance between the two SS’s; iris area: the cumulative cross-sectional area of the full length (from spur to pupil) of the iris; CSD: the distance between the two horizontal diameters for the ciliary sulcus.

**Figure 2 jcm-12-00327-f002:**
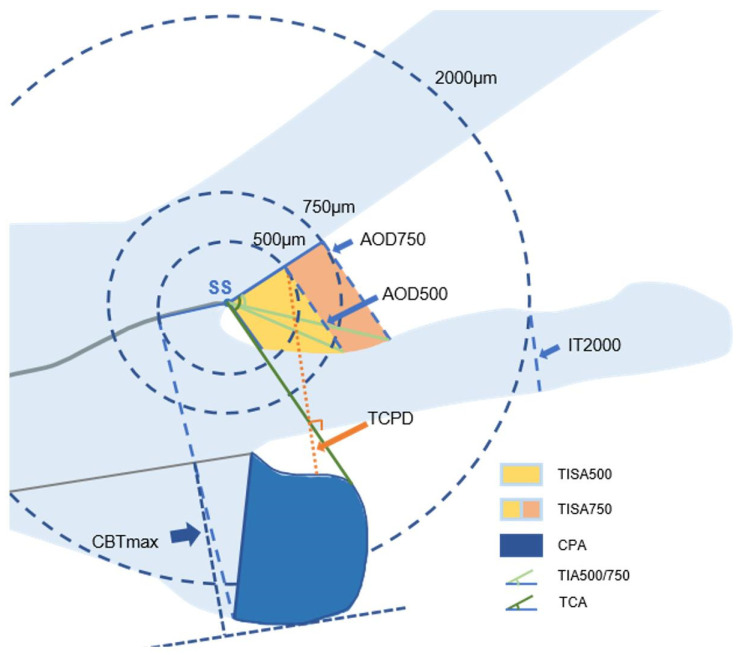
Anterior segment parameters on the radial scans at the superior, nasal, inferior, and temporal positions measured by ultrasound biomicroscopy (UBM). (Illustrator: K.Y.). IT 750/2000: iris thickness of 750 μm and 2000 μm from the SS; AOD 500/750: the distance between the posterior corneal surface and the anterior iris surface on a line perpendicular to the trabecular meshwork at 500 μm and 750 μm from the SS; TISA 500/750: the area bounded anteriorly by AOD 500 and AOD 750, as determined posteriorly by a line drawn from the SS perpendicular to the plane of the inner scleral wall to the iris, superiorly by the inner corneoscleral wall, and inferiorly by the iris surface; TIA 500/750: the apex of the angle at the iris recess and the arms of the angle passing through a point on the trabecular meshwork at 500 μm and 750 μm from the SS and the point on the iris perpendicularly opposite; CPA: the cross-sectional area of the ciliary process bounded laterally by a line connecting the insertion location of the iris into the ciliary body and the crosspoint of a line of 500 μm from the SS perpendicular to the plane of the inner scleral wall to the ciliary process, and internally by the ciliary process surface; TCA: the angle between the posterior corneal surface and the anterior surface of the ciliary body; TCPD: the length of the line extending from the corneal endothelium 500 mm from the SS perpendicularly through the posterior surface of the iris to the ciliary process; CBTmax: the thickest location of the ciliary body.

**Figure 3 jcm-12-00327-f003:**
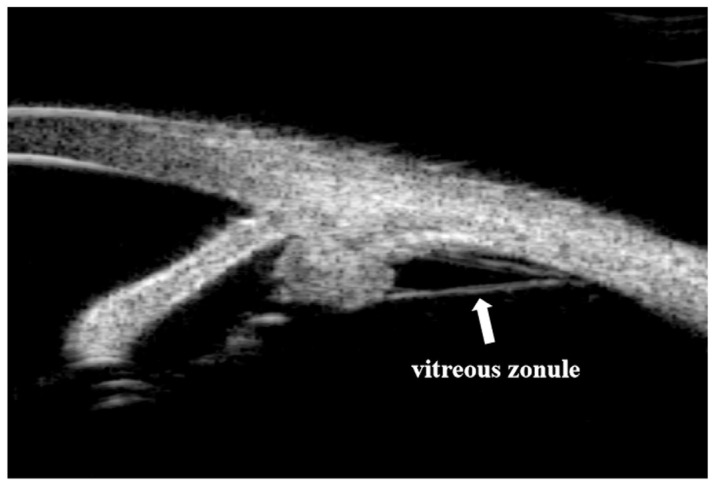
Vitreous zonule (arrows) visible in an ultrasound biomicroscopy (UBM) image of a 63-year-old acute PACG female.

**Table 1 jcm-12-00327-t001:** Comparison of demographics, clinical characteristics, and preoperative anterior segment UBM parameters among the three groups.

	Chronic PACG	Acute PACG	Open-Angle Cataract	*p*-Value *
Eyes	22	22	22	
Demographics and clinical characteristics				
Gender (male/female)	9/13	6/16	5/17	0.105
Age (years)	66.55 ± 7.07	70.73 ± 8.77	71.00 ± 7.03	0.394
BCVA (decimal)	0.38 ± 0.23	0.28 ± 0.30	0.35 ± 0.19	0.353
IOP (mmHg)	20.23 ± 9.16	23.68 ± 15.66	13.75 ± 2.09	0.001 *
Axial length (mm)	22.40 ± 0.65	22.58 ± 1.19	24.31 ± 1.79	0.000 *
Preoperative anterior segment parameters on UBM				
Preoperative CPA (mm^2^)	0.60 ± 0.20	0.62 ± 0.17	0.41 ± 0.16	0.014 *
Preoperative TCA (degree)	54.98 ± 12.03	49.02 ± 11.04	83.94 ± 15.36	0.000 *
Preoperative TCPD (mm)	0.55 ± 0.14	0.50 ± 0.14	0.80 ± 0.18	0.000 *
Preoperative CBTmax (mm)	1.17 ± 0.11	1.16 ± 0.11	1.31 ± 0.12	0.000 *
Preoperative CSD (mm)	10.37 ± 0.63	10.42 ± 0.53	10.83 ± 0.98	0.069
Preoperative AQD (mm)	1.93 ± 0.23	1.56 ± 0.23	2.76 ± 0.41	0.000 *
Preoperative ACW (mm)	11.27 ± 0.48	11.30 ± 0.63	11.58 ± 0.75	0.020 *
Preoperative LV (mm)	0.84 ± 0.30	1.13 ± 0.25	0.34 ± 0.24	0.000 *
Preoperative PD (mm)	2.33 ± 0.61	4.04 ± 1.38	2.71 ± 0.54	0.007 *
Preoperative IT 750 (µm)	0.40 ± 0.09	0.39 ± 0.07	0.48 ± 0.06	0.024 *
Preoperative IT 2000 (µm)	0.45 ± 0.07	0.49 ± 0.10	0.52 ± 0.06	0.109
Preoperative TIA 500 (degree)	3.67 ± 4.48	2.57 ± 5.66	30.15 ± 11.18	0.000 *
Preoperative TIA 750 (degree)	3.69 ± 4.39	2.57 ± 5.78	29.23 ± 10.16	0.000 *
Preoperative AOD 500 (µm)	0.03 ± 0.04	0.02 ± 0.05	0.32 ± 0.15	0.000 *
Preoperative AOD 750 (µm)	0.05 ± 0.06	0.04 ± 0.08	0.45 ± 0.20	0.000 *
Preoperative TISA 500 (mm^2^)	0.01 ± 0.01	0.01 ± 0.02	0.11 ± 0.06	0.000 *
Preoperative TISA 750 (mm^2^)	0.02 ± 0.02	0.02 ± 0.04	0.21 ± 0.10	0.000 *
Preoperative iris area (mm^2^)	1.92 ± 0.17	1.68 ± 0.37	2.16 ± 0.28	0.002 *
Preoperative number of quadrants of VZ	1.55 ± 1.26	2.00 ± 1.07	2.73 ± 1.07	0.007 *

PACG = primary angle-closure glaucoma; BCVA = best corrected visual acuity; IOP = intraocular pressure; CPA = ciliary process area; TCA = trabecular-ciliary angle; TCPD = trabecular-ciliary process distance; CBTmax = maximum ciliary body thickness; CSD = ciliary sulcus diameter; AQD = aqueous depth; ACW = anterior chamber width; LV = lens vault; PD = pupil diameter; IT 750 = iris thickness at 750 µm; IT 2000 = iris thickness at 2000 µm; TIA 500 = trabecular iris angle at 500 µm; TIA 750 = trabecular iris angle at 750 µm; AOD 500 = angle-opening distance at 500 µm; AOD 750 = angle-opening distance at 750 µm; TISA 500 = trabecular iris space area at 500 µm; TISA 750 = trabecular iris space area at 750 µm; VZ= vitreous zonule. * χ2 tests were used to compare gender; analysis of variance (ANOVA) tests or Kruskal–Wallis tests were used to compare age, AQD, PD, LV, ACW, CSD, iris area, IT 750/2000, AOD 500/750, TISA 500/750, TIA 500/750, CPA, TCA, TCPD, CBTmax, and the number of quadrants of VZ (*n* = 22).

**Table 2 jcm-12-00327-t002:** The comparisons of the changes in the anterior segment parameters after cataract surgery among the three groups.

Parameters	Chronic PACG	Acute PACG	Open-Angle Cataract	*p*-Value *
ΔCPA (mm^2^)	−0.03 ± 0.15	−0.10 ± 0.17	−0.00 ± 0.12	0.182
ΔTCA (degree)	8.70 ± 9.57	12.38 ± 9.56	6.02 ± 7.92	0.096
ΔTCPD (mm)	0.05 ± 0.11 (*n* = 17)	0.04 ± 0.13 (*n* = 16)	0.05 ± 0.13 (*n* = 15)	0.904
ΔCBTmax (mm)	0.07 ± 0.14	0.06 ± 0.08	0.12 ± 0.12	0.155
ΔCSD (mm)	−0.23 ± 0.55	−0.29 ± 0.42	−0.17 ± 0.53	0.715
ΔAQD (mm)	1.45 ± 0.27	1.79 ± 0.43	1.37 ± 0.40	0.000 *
ΔACW (mm)	0.34 ± 0.51	0.12 ± 0.49	0.14 ± 0.43	0.204
ΔLV (mm)	−1.22 ± 0.31	−1.62 ± 0.35	−1.26 ± 0.35	0.000 *
ΔPD (mm)	−0.21 ± 0.64	−0.06 ± 0.94	−0.61 ± 0.58	0.054
ΔIT 750 (µm)	0.03 ± 0.05	0.07 ± 0.07	−0.01 ± 0.06	0.006 *
ΔIT 2000 (µm)	0.03 ± 0.05	0.08 ± 0.08	0.02 ± 0.06	0.004 *
ΔTIA 500 (degree)	13.41 ± 11.76	6.90 ± 9.22	11.33 ± 8.74	0.362
ΔTIA 750 (degree)	14.05 ± 11.33	7.19 ± 9.22	12.09 ± 8.32	0.264
ΔAOD 500 (µm)	0.14 ± 0.13	0.07 ± 0.10	0.14 ± 0.11	0.101
ΔAOD 750 (µm)	0.22 ± 0.18	0.11 ± 0.14	0.22 ± 0.16	0.082
ΔTISA 500 (mm^2^)	0.05 ± 0.04	0.02 ± 0.03	0.05 ± 0.04	0.103
ΔTISA 750 (mm^2^)	0.09 ± 0.07	0.04 ± 0.06	0.10 ± 0.07	0.053
ΔIris area (mm^2^)	0.21 ± 0.22	0.10 ± 0.28	0.09 ± 0.22	0.440
ΔNumber of quadrants of VZ	−0.73 ± 1.08	−0.82 ± 0.96	−1.27 ± 0.77	0.063

PACG = primary angle-closure glaucoma; Δ represents the difference between pre- and postoperative parameters (postoperative minus preoperative parameters); CPA = ciliary process area; TCA = trabecular-ciliary angle; TCPD = trabecular-ciliary process distance; CBTmax = maximum ciliary body thickness; CSD = ciliary sulcus diameter; AQD = aqueous depth; ACW = anterior chamber width; LV = lens vault; PD = pupil diameter; IT 750 = iris thickness at 750 µm; IT 2000 = iris thickness at 2000 µm; TIA 500 = trabecular iris angle at 500 µm; TIA 750 = trabecular iris angle at 750 µm; AOD 500 = angle-opening distance at 500 µm; AOD 750 = angle-opening distance at 750 µm; TISA 500 = trabecular iris space area at 500 µm; TISA 750 = trabecular iris space area at 750 µm; VZ = vitreous zonule. *Analysis of variance (ANOVA) tests or Kruskal–Wallis one-way ANOVA (k samples) tests (*n* = 22).

**Table 3 jcm-12-00327-t003:** Intra-observer and inter-observer intra-class coefficients of the preoperative and postoperative parameters.

Parameters	Intra-Class Coefficients	
	Intra-Observer	Inter-Observer
Preoperative CPA (mm^2^)	0.956	0.872
Postoperative CPA (mm^2^)	0.913	0.852
Preoperative TCA (degree)	0.969	0.862
Postoperative TCA (degree)	0.982	0.927
Preoperative TCPD (mm)	0.967	0.869
Postoperative TCPD (mm)	0.968	0.914
Preoperative CBTmax (mm)	0.983	0.909
Postoperative CBTmax (mm)	0.970	0.876
Preoperative CSD (mm)	0.980	0.953
Postoperative CSD (mm)	0.973	0.980
Preoperative AQD (mm)	0.999	0.971
Postoperative AQD (mm)	0.999	0.997
Preoperative ACW (mm)	0.923	0.872
Postoperative ACW (mm)	0.916	0.873
Preoperative LV (mm)	0.968	0.933
Postoperative LV (mm)	0.950	0.885
Preoperative PD (mm)	0.998	0.988
Postoperative PD (mm)	0.988	0.963
Preoperative IT 750 (µm)	0.949	0.909
Postoperative IT 750 (µm)	0.906	0.873
Preoperative IT 2000 (µm)	0.913	0.876
Postoperative IT 2000 (µm)	0.975	0.934
Preoperative TIA 500 (degree)	0.974	0.940
Postoperative TIA 500 (degree)	0.988	0.954
Preoperative TIA 750 (degree)	0.974	0.966
Postoperative TIA 750 (degree)	0.983	0.965
Preoperative AOD 500 (µm)	0.965	0.936
Postoperative AOD 500 (µm)	0.988	0.946
Preoperative AOD 750 (µm)	0.963	0.962
Postoperative AOD 750 (µm)	0.985	0.972
Preoperative TISA 500 (mm^2^)	0.976	0.947
Postoperative TISA 500 (mm^2^)	0.992	0.984
Preoperative TISA 750 (mm^2^)	0.963	0.965
Postoperative TISA 750 (mm^2^)	0.986	0.976
Preoperative iris area (mm^2^)	0.991	0.910
Postoperative iris area (mm^2^)	0.964	0.887
Preoperative number of quadrants of VZ	0.869	0.862
Postoperative number of quadrants of VZ	0.904	0.865

CPA = ciliary process area; TCA = trabecular-ciliary angle; TCPD = trabecular-ciliary process distance; CBTmax = maximum ciliary body thickness; CSD = ciliary sulcus diameter; AQD = aqueous depth; ACW = anterior chamber width; LV = lens vault; PD = pupil diameter; IT 750 = iris thickness at 750 μm; IT 2000 = iris thickness at 2000 μm; TIA 500 = trabecular iris angle at 500 μm; TIA 750 = trabecular iris angle at 750 μm; AOD 500 = angle-opening distance at 500 μm; AOD 750 = angle-opening distance at 750 μm; TISA 500 = trabecular iris space area at 500 μm; TISA 750 = trabecular iris space area at 750 μm; VZ = vitreous zonule.

## Data Availability

Data presented in this study are available from the corresponding author upon request.

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
