# Peer review of "Anterior Segment Parameter Changes after Cataract Surgery in Open-Angle and Angle-Closure Eyes: A Prospective Study"

_jcm, 2022, doi:10.3390/jcm12010327_

Round 1

Reviewer 1 Report

General comments:

It was a very interesting study with many important parameters measured. I congratulate the authors on a good and precise study.

Specific comments

Line 73: The study included a total of 68 eyes in three groups. Did the authors perform a power calculation of how many eyes should be included?

Line 132: Iris thickness was measured at a radius of 750 µm and 2000 µm from the scleral spur. Can a brief explanation be given here as to why this particular distance was chosen?

Line 180: Were all patients implanted with the same IOL? Which IOL is it? This should be briefly declared here, especially whether the IOL is an acrylic material and whether it is hydrophobic or hydrophilic.

Line 185: Can the concentration of the hyaluronic acid still be listed?

Line 193: Why was the patient prescribed oxybuprocaine postoperatively for one week? Oxybuprocaine is epithelial toxic. A little explanation would be good.

Line 208: How was it calculated whether a data set is further calculated with parametric or non-parametric test?  How were the data sets analysed?

Line 368: It would be good if, in addition to the effects, the clinical relevance could be discussed more prominently as a consequence in everyday clinical practice or as a possible therapeutic importance.

Author Response

Point 1: Line 73: The study included a total of 68 eyes in three groups. Did the authors perform a power calculation of how many eyes should be included?

Response 1: We thank the reviewer for this comment. Sample size of eyes was estimated based on the One-Way Analysis of Variance F-Tests using statistical sample size software PASS 15.0 (NCSS, Kaysville, UT, United States). Based on the initial results from our study of about 10 eyes per group, typical parameters (trabecular-ciliary angle, maximum ciliary body thickness, trabecular-ciliary process distance) were chosen to make sure that the sample size could support the variables. We calculated the required effect size given our sample of N = 60 (and alpha: 0.017, power: 0.9), which is similar to the sample size included in our study. In addition, the purpose of our study was to evaluate how preoperative and postoperative anterior segment parameters change after cataract surgery in the same eyes. Paired contrast t-tests were used for the preoperative and postoperative comparison Sample size of eyes was estimated based on the Tests for Paired Means using statistical sample size software PASS 15. We calculated the required effect size given our sample of N = 36 (and alpha: 0.05, power: 0.9) through the initial results, which is smaller than the sample size included in our study. In addition, Ünsal et al. [1] evaluated morphologic changes in the anterior segment using ultrasound biomicroscopic (UBM) imaging after phacoemulsification and intraocular lens implantation in 36 eyes. Twenty-six eyes had clear lens extraction by phacoemulsification, while 24 eyes underwent trabeculectomy to document the anatomical effects of clear lens extraction on anterior chamber angle in patients [2]. We included a larger sample of trials than similar investigations across previous clinical disciplines [1,2]. Although the sample size has been calculated, small sample size is indeed one of the limitations. We also mentioned this in the limitations section at the end of the article.

Point 2: Line 132: Iris thickness was measured at a radius of 750 µm and 2000 µm from the scleral spur. Can a brief explanation be given here as to why this particular distance was chosen?

Response 2: We thank the reviewer for raising this question. In previous studies [3-5], UBM and anterior segment optical coherence tomography have utilised iris thickness at 750 µm (IT 750) and at 2000 µm (IT 2000) to represent the iris thickness, which demonstrated that an increased iris thickness was associated with narrow angles and angle closure. In addition, we proposed in our previous study that, in primary angle-closure eyes, the thinner the IT 750, the greater the risk of an acute attack [5]. As an important indicator to evaluate the state of the iris, we used IT 750 and IT 2000 as observation parameters. Although there have been many studies confirming that these two parameters can be used to evaluate iris status, we also acknowledge that utilizing only two locations may not be ideal, as they may not be representative of the entire iris. Hence, further explorations about iris thickness and configuration are warranted.

Point 3: Line 180: Were all patients implanted with the same IOL? Which IOL is it? This should be briefly declared here, especially whether the IOL is an acrylic material and whether it is hydrophobic or hydrophilic.

Response 3: We thank the reviewer for this valuable suggestion. All the patient implanted with the same hydrophobic acrylic TECNIS ZA9003 IOL (Abbott Medical Optics Inc., Santa Ana, CA, USA.) and we added the details in Page 5, Line 182-183.

Point 4: Line 185: Can the concentration of the hyaluronic acid still be listed?

Response 4: We thank the reviewer for pointing this out. The concentration of the hyaluronic acid was 1.7% and we added the details in Page 6, Line 188.

Point 5: Line 193: Why was the patient prescribed oxybuprocaine postoperatively for one week? Oxybuprocaine is epithelial toxic. A little explanation would be good.

Response 5: We thank the reviewer for pointing this out. It is our negligence to write the wrong name of the eye drop. We used 0.4% oxybuprocaine hydrochloride (Benoxil®; Santen, Tokyo, Japan) eye drops for topical anesthesia before and at 3 months after cataract surgery when performing UBM evaluation. And we used 0.5% levofloxacin (Cravit®; Santen, Osaka, Japan) eye drops four times a day for 1 week after surgery to prevent bacterial infections. We have corrected this error in Page 3, Line 107-108 and Page 6, Line 195-196, respectively. Many thanks for pointing out the mistake again.

Point 6: Line 208: How was it calculated whether a data set is further calculated with parametric or non-parametric test?  How were the data sets analysed?

Response 6: We thank the reviewer for pointing this out. The Shapiro-Wilk test was used to test whether the variables conform to normal distribution. The test showed that the data (age, CPA, TCPD, AQD, CSD, ACW, LV, IT 750 and iris area) conforming to normal distribution; therefore, analysis of Variance (ANOVA) tests was used to compare parametric variables. Alternatively, Kruskal-Wallis test in case of non-normally distributed variables was used to compare non-parametric variables (BCVA, IOP, axial length, TCA, CBTmax, PD, IT 2000, AOD 500/750, TIA 500/750, TISA 500/750 and number of quadrants of VZ). We have added a description of the analytical method in Page 6, Line 210-216.

Point 7: Line 368: It would be good if, in addition to the effects, the clinical relevance could be discussed more prominently as a consequence in everyday clinical practice or as a possible therapeutic importance.

Response 7: We thank the reviewer for this valuable suggestion. In the current study, we showed that as a natural anatomical abnormality, the anterior rotation of ciliary processes in primary angle-closure glaucoma (PACG) patients may not be associated with the lens factor. It is important to increase our understanding of the pathogenesis of PACG, to urge us to further explore the cause of anterior rotation of ciliary processes such as the association with the presence of vitreous zonule. Reducing the volume of the ciliary body or relieving the anterior rotation of ciliary processes may further help us to carry out new treatments for PACG patients, such as cyclophotocoagulation. We have added this in the discussion section (Page 11, Line 376-382).

Thank you again for your positive comments and valuable suggestions to improve the quality of our manuscript. After carefully studying your comments, we tried our best to incorporate the corresponding revisions and we truly hope that it will meet with your expectations and approval. If there are any other modifications we could make, we would like to modify them.

References

  1. Ünsal, E.; Eltutar, K.; Muftuoglu, Ä°.K. Morphologic Changes in the Anterior Segment using Ultrasound Biomicroscopy after Cataract Surgery and Intraocular Lens Implantation. Eur J Ophthalmol 2016, 27, 31-38, doi:10.5301/ejo.5000812.
  2. Man, X.; Chan, N.C.Y.; Baig, N.; Kwong, Y.Y.Y.; Leung, D.Y.L.; Li, F.C.H.; Tham, C.C.Y. Anatomical effects of clear lens extraction by phacoemulsification versus trabeculectomy on anterior chamber drainage angle in primary angle-closure glaucoma (PACG) patients. Graefes Arch Clin Exp Ophthalmol 2015, 253, 773-778, doi:10.1007/s00417-015-2936-z.
  3. Wang, B.; Sakata, L.M.; Friedman, D.S.; Chan, Y.H.; He, M.; Lavanya, R.; Wong, T.Y.; Aung, T. Quantitative iris parameters and association with narrow angles. Ophthalmology 2010, 117, 11-17, doi:10.1016/j.ophtha.2009.06.017.
  4. Wang, B.S.; Narayanaswamy, A.; Amerasinghe, N.; Zheng, C.; He, M.; Chan, Y.H.; Nongpiur, M.E.; Friedman, D.S.; Aung, T. Increased iris thickness and association with primary angle closure glaucoma. Br J Ophthalmol 2011, 95, 46-50, doi:10.1136/bjo.2009.178129.
  5. You, S.; Liang, Z.; Yang, K.; Zhang, Y.; Oatts, J.; Han, Y.; Wu, H. Novel Discoveries of Anterior Segment Parameters in Fellow Eyes of Acute Primary Angle Closure and Chronic Primary Angle Closure Glaucoma. Invest Ophthalmol Vis Sci 2021, 62, 6, doi:10.1167/iovs.62.14.6.

Reviewer 2 Report

The present manuscript entitled "Anterior segment parameter changes after cataract surgery in open angle and angle-closure eyes: A prospective study" by Yang et al is focused on to investigate the anterior rotation of ciliary body after cataract surgery in PACG and normal eyes. The study was well planned and written well. However, there are few concerns that need to be addressed.

The definition of ACD is not correct in the present study. Click the link below to confirm the definition. According to the link below, the authors should use Aqueous depth (AQD) instead of ACD.

https://www.aaojournal.org/action/showPdf?pii=S0161-6420%2811%2900247-8

The authors used words “Natural anatomical abnormality” in the conclusion. What is natural anatomical abnormality? Are there any parameters reflecting the natural anatomical abnormality in the parameters used in the present study?

Author Response

Response to Reviewer 2 Comments

Point 1: The definition of ACD is not correct in the present study. Click the link below to confirm the definition. According to the link below, the authors should use Aqueous depth (AQD) instead of ACD. https://www.aaojournal.org/action/showPdf?pii=S0161-6420%2811%2900247-8

Response 1: We thank the reviewer for pointing this out. It is our negligence to write the confusion of the definition. According to your kind suggestion and previous study [1], We adopted the suggested (correct) definition about aqueous depth (AQD). We defined AQD as “the distance between the posterior surface of the central cornea and the anterior surface of the crystalline lens or IOL in the midline of the pupil” in Page 3, Line 123-125. It is unfortunate that this confusion of the definition has persisted and this problem also existed in the research we cited [2]. For a better understanding, we have replaced all anterior chamber depths with AQDs in red.

Point 2: The authors used words “Natural anatomical abnormality” in the conclusion. What is natural anatomical abnormality? Are there any parameters reflecting the natural anatomical abnormality in the parameters used in the present study?

Response 2: We thank the reviewer for this comment. “Natural anatomical abnormality” refers to the fact that the anterior rotation of ciliary processes in primary angle-closure glaucoma (PACG) patients may not be associated with the lens factor. Trabecular-ciliary process distance (TCPD) is the length of the line extending from the corneal endothelium 500mm from the scleral spur perpendicularly through the posterior surface of iris to the ciliary process.; Trabecular-ciliary angle (TCA) is the angle between the posterior corneal surface and the anterior surface of the ciliary body. He et al. [3] proposed that healthy Chinese had smaller TCPD, TCA and thinner ciliary body thickness than healthy Caucasians. They suggested that the Chinese have a more anteriorly positioned lens, which may contribute to more anteriorly positioned ciliary processes and the thinner ciliary body might cause anterior positioning and thickening of the lens through the greater anterior positioning of the ciliary processes and loosening of the zonules [3]. The result of the present study was no significant TCPD increase in all groups postoperatively, which did not support the hypothesis of anterior rotation of ciliary processes resulting from anterior traction of crystalline lens. In addition to TCPD, TCA could also reflect the anterior positioning of the ciliary processes. In the present study, the changes in TCA and TCPD among open angle eyes and different subtypes of PACG eyes were not statistically significant postoperatively, which indicated that the attenuate effects of lens extraction were consistent in all three groups. Therefore, the changes in TCPD and TCA postoperatively reflects that the anterior rotation of ciliary processes might be a natural anatomical structure in PACG patients. Explanations can be found in the discussion section in detail.

Thank you again for your positive comments and valuable suggestions to improve the quality of our manuscript. If there are any other modifications we could make, we would like to modify them, and we really appreciate your help.

References

  1. Ünsal, E.; Eltutar, K.; Muftuoglu, Ä°.K. Morphologic Changes in the Anterior Segment using Ultrasound Biomicroscopy after Cataract Surgery and Intraocular Lens Implantation. Eur J Ophthalmol 2016, 27, 31-38, doi:10.5301/ejo.5000812.
  2. Man, X.; Chan, N.C.Y.; Baig, N.; Kwong, Y.Y.Y.; Leung, D.Y.L.; Li, F.C.H.; Tham, C.C.Y. Anatomical effects of clear lens extraction by phacoemulsification versus trabeculectomy on anterior chamber drainage angle in primary angle-closure glaucoma (PACG) patients. Graefes Arch Clin Exp Ophthalmol 2015, 253, 773-778, doi:10.1007/s00417-015-2936-z.
  3. He, N.; Wu, L.; Qi, M.; He, M.; Lin, S.; Wang, X.; Yang, F.; Fan, X. Comparison of Ciliary Body Anatomy between American Caucasians and Ethnic Chinese Using Ultrasound Biomicroscopy. Curr Eye Res 2015, 41, 485–491, doi:10.3109/02713683.2015.1024869.